materials science/environmental chemistry/green chemistry

adsorption, sisal-based activated carbon fibre, butane working capacity, fuel evaporation, Thomas model, Clark model

**Author for correspondence:**
Hailong Li
e-mail: felhl@scut.edu.cn

This article has been edited by the Royal Society of Chemistry, including the commissioning, peer review process and editorial aspects up to the point of acceptance.

[†]Present address: School of Light Industry and Engineering, South China University of Technology, Guangzhou 510641, People's Republic of China.

# Sisal xylem fibre-based activated carbon fibres for fuel adsorption: effect of thermal stabilization of diammonium phosphate

## Zhuo Deng, Jian Hu and Hailong Li[†]

School of Light Industry and Engineering, South China University of Technology, Guangzhou 510641, People's Republic of China

ZD, 0000-0002-5388-5653; HL, 0000-0002-4665-5921

Activated carbon fibres (ACFs) are considered as the next generation of activated carbon products. However, lack of structural diversity in pore structure and high prices of raw materials for ACFs has restrained the development of ACF materials. In this paper, a sisal-based activated carbon fibre (SACF) material was prepared from sisal wastes with a unique thermal stabilization treatment to maintain fibrous shapes of SACFs while dispersing in solutions, and the SACFs were prepared as raw fibre materials for fuel evaporation emissions controlling products. Experimental results of $N_2$ adsorption showed that SACF has a typical I-type adsorption isotherm, with specific surface area of SACF samples of approximately 1200 $m^2\,g^{-1}$, and mainly microporous pore structure. Compared with commercial samples (specific surface area, 1841.29 $m^2\,g^{-1}$), the butane working capacities of SACF for advanced fuel evaporation emissions controlling product was 0.4 g/100 ml higher. Furthermore, two dynamic models, Thomas model and Clark model, were applied to adsorption breakthrough data, which showed excellent fit. And it indicated from the adsorption breakthrough curves and parameters of both models that the SACF has better performance in fuel adsorption and desorption process than the commercial samples.

## 1. Introduction

Volatile fuel leaked from vehicles is among the most commonly occurring and widely distributed contaminants in the environment. Evaporative fuel is composed of a homogeneous mixture of small, relatively lightweight hydrocarbons (HCs; mainly $C_4$–$C_{12}$). More than 15 chemical components in fuel are

**Figure 1.** (*a*) Drying sisal ribbon fibres; (*b*) sisal xylem fibres.

considered as hazardous chemicals. There are strict rules made by environmental agencies regulating emission of hydrocarbon [1,2].

Activated carbon (AC) has been widely used in evaporation emission control system, such as carbon tank–fuel tank system, for removal of volatile HC. The high-level porosity and adsorptive capacity provide the adsorption capacity and recovery of useful substances [3–5]. With the increasingly stringent emission standards, the automotive industry has higher requirements for pollution control. Traditional AC materials are generally used to control fuel evaporation emissions, but due to its own defects such as low adsorption capacity, slow adsorption speed and easy peeling of particles, the new generation of AC products, activated carbon fibres (ACFs), have attracted more attention today [3,6].

Recent studies show that ACF is considered as an excellent HC adsorbent with better performance than other classical adsorbent, such as granular activated carbon (GAC), under dynamic adsorption processes [4,7,8]. Except for the expensive man-made fibres, the natural fibres made up mostly of cellulose and hemicellulose could be considered as excellent precursors of ACFs. Sisal hemp, an important crop in Mexico, can provide applicable precursor of ACF [7–9]. Compared with man-made fibres, like polyacrylonitrile (PAN) fibres, sisal fibres is not only competitive in mechanicalproperties (tensile strength, Young's modulus, etc.), but also more prominent in economy and environmental protection [10,11]. And lignin content of sisal fibres (8%) is lower than many plant fibres, which makes sisal fibres easily separate with alkali treatment and remain fibre structure through activation process [10].

Statistics reveal that global sisal ribbon fibre production amounted to 280 kt and China produced 38 kt in 2017. The ribbon fibres extracted from sisal leaves have fine mechanical strength and modulus properties compared with glass fibres [12,13]. But in this work, we used sisal xylem fibres as precursors instead of 'fancy' ribbon fibres. The xylem fibres which are usually considered as the useless part of sisal leaves have irregular shape and easily break up during processing. It was reckoned the xylem fibre production amounted to 0.7t/t sisal ribbon fibre [14–16]. These fibres are inferior in mechanical properties, but they have a similar constituent as ribbon fibres: 65% cellulose and 22% hemicellulose. It indicates that most parts of the 'useless' fibres are good precursors of ACFs. There are many articles detailing studies producing ACFs from all kinds of plants [7–9,17,18]. Remarkable properties were investigated in those articles, such as high specific surface area, unique pore structure and rapid adsorption rate, but the shape of ACFs after high-temperature activation processing was not. In those studies, most ACFs produced from natural fibres melted together, which were no different from AC. Without steady mechanical properties, ACFs are easily broken up. Therefore, we found a unique thermal stabilization measure to maintain the fibre shape of ACF, on the premise that ACF also has good fuel adsorption performance, which was studied in a dynamic adsorption system. And ACFs we made showed superior performance compared with advanced commercial AC.

## 2. Material and methods

### 2.1. Raw materials

In this paper, sisal xylem fibres (the raw materials) were produced from some parts of sisal which were collected from a farm located in Fusui Town, Guangxi Province, PRC (figure 1*a*). The diameter of xylem fibres (figure 1*b*) were in the range of 0.2–1.3 mm.

To prepare the precursors, non-fibrous components were removed from sisal xylem fibres by mechanical compression in a centrifugal separator. Prepared alkaline solution with NaOH solution with concentrations of 12%, $Na_2SO_3$ solution with concentrations of 3% and anthraquinones solution

with concentrations of 0.06%. Mixed with the alkaline solution, remaining fibres were heated to 150°C in a digester for 3.5 h. After removing the residual alkaline solution with distilled water, the sisal fibres (precursor of sisal-based ACF) were extracted through a clean and dry process.

## 2.2. Thermal stabilization and chemical activation

Typical activation methods include treating carbon sources with oxidizing gas or chemical impregnant in a high-temperature inert atmosphere. Sisal fibres melt or burn off significantly during the activation process, therefore thermal stabilization is necessary. The detailed preparation process consists of two parts.

(1) Thermal stabilization process: two groups of sisal fibres were respectively impregnated with $(NH_4)_2HPO_4$ (DAP, diammonium phosphate) solution with concentrations of 20% (labelled as SACF-DAP20) and 40% (labelled as SACF-DAP40). In addition, some sisal fibres were soaked with distilled water as blank samples (labelled as SACF-DAP0). After standing for 12 h, the samples were heated to 180°C for 1 h in a ventilated oven.
(2) Chemical activation process: each sample was impregnated with $ZnCl_2$ solution with a mass concentration of 30%. After standing for 12 h, the samples were heated to 750°C in a pipe furnace flushed with nitrogen for 1 h and natural cooling under a nitrogen atmosphere. The samples were then washed several times first with dilute hydrochloric acid and then with distilled water until the pH turned neutral. After drying in the ventilated oven at 120°C, the samples of sisal-based ACFs were obtained.

## 2.3. Activated carbon fibres characterization

The morphologies of ACFs were observed using a scanning electron microscope (Phenom, US) at an operation voltage of 5 kV. A transmission electron microscope (JEOL, JEM-2200FS, Japan) was used to further characterize the crystallographic feature of the prepared samples at an operation voltage of 200 kV.

The pore characteristics of samples were investigated with nitrogen adsorption volume method using a surface area and porosity analyser (Micromeritics, ASAP 2460, US). The samples were outgassed for 12 h under a vacuum at 300°C before the analysis. The nitrogen adsorption–desorption isotherms were obtained by the analyser to calculate BET specific surface area ($S_{BET}$), micropore surface area ($S_{MIC}$), mesopore surface area ($S_{MES}$), micropore volume ($V_{MIC}$), mesopore volume ($V_{MES}$) with t-pot and BET methods and mean micropore width ($W_{MIC}$) with Horvath–Kawazoe method. The pore size distribution (PSD) was investigated by the nonlinear density functional theory (NLDFT).

## 2.4. Fuel adsorption

Fuel adsorption performance was evaluated by butane working capacity (BWC) testing referring to ASTM D5228-92 (2015). BWC is an adsorption indicator to show the reversible adsorption performance of butane [5,19]. This test is especially suitable for the adsorbent efficiency of processes such as evaporation control devices in which the adsorbate is trapped and recovered in a ring system. The BWC provides important adsorption information to reflect the character and evolution of the porosity of the adsorbent. Micropores less than 2 nm in size absorb butane due to overlapping potential fields in adjacent walls but retain it strongly. The size of mesopores generally varies between 2 and 50 nm, which are not an appropriate size to capture small molecules, but big enough to desorb butane easily.

The samples were tested in a constant shaped tube as determined in ASTM D2854 standard procedure. The tube was then kept in a 25°C water bath and the butane was allowed to flow at 150 ml min$^{-1}$ for 10 min. After being weighed, the samples continued butane adsorption process for an additional 5 min. After reaching a relatively stable sample weight, desorption was affected by the passage of clean dry air through the test system. The BWC is calculated by the difference in mass between the saturated adsorbent and the desorbed adsorbent.

Also the concentration of butane leaked through the outlet end was diluted and detected by a photoionization detector (RAE, PGM 7300, US).

**Table 1.** Thermal stabilization by DAP of SACF samples.

| samples | thermal stabilizer | temperature (°C) | burn-off (%) |
|---|---|---|---|
| SACF-DAP0 | DI Water | 180 | 86.2 |
| SACF-DAP20 | 20% DAP | 180 | 74.3 |
| SACF-DAP40 | 40% DAP | 180 | 64.5 |

# 3. Results and discussion

## 3.1. Burn-off

$(NH_3)_2HPO_4$ (DAP) is usually used as a phosphate flame retardant or an activator of AC preparation. In this study, DAP was used as a thermal stabilizer to keep the original fibre shape of sisal fibres after chemical activation [20,21].

An activation catalyser is conclusive for pore structure generation of sisal fibres, so $ZnCl_2$ was used as the chemical activation catalyser.

The burn-off of sisal-based ACF is used to analyse the effect of thermal stabilization on pore development. The equation to calculate the burn-off is as follows:

$$\text{burn-off} \ (\%) = \frac{m_i - m_f}{m_i} \times 100\%, \tag{3.1}$$

where $m_i$ is the initial mass of precursor (g) and $m_f$ is final mass of ACF (g) respectively.

The burn-off for SACF-DAP0, SACF-DAP20 and SACF-DAP40 samples are 86.2%, 74.3% and 64.5%, respectively. The increment of burn-off usually suggests enrichment of carbon and development in the pore structure of ACF. The development of the porosity has main contribution to increment of ACF specific surface area. But with the protection of a thermal stabilizer, SACF maintained enhanced pore structure with a higher yield. A summary of the heat-treatment processes employed are listed in table 1. Burn-off decreased with the increase of DAP impregnation ratio, which revealed that DAP as thermal stabilizer can effectively suppress the thermal loss of the carbonization process.

## 3.2. Morphological characterization

The morphologies of the ACF samples were evaluated by the SEM images. Typical images of sisal samples are shown in figure 2.

Figure 2a shows the sisal, the raw material, with pore structure having typical characteristics of hemp plants which contained bundle sisal fibres and parenchyma cells. Diameters of sisal xylem fibres were in the range of 150–300 μm. There were porous channels with inner diameter of approximately 10 μm, which are usually considered as effective pores generation sites in many article papers. But parenchyma cells among the fibres are non-fibre elements which have a negative effect on activation process of ACF preparation.

Heated in a digester, the pectin and lignin, binder of fibres, in fibre bundle will be massively dissolved in alkaline solution. And the sisal fibres will be well separated as shown in figure 2b. The specific surface area increased after the separation, which made fibres more thoroughly with chemical impregnation and pores more generated on the surface. The longitudinal fibres with rough appearance were characteristic of natural plant fibres. The diameter of sisal fibres were mostly 7–15 μm. Typical images of the ACFs prepared from sisal ACF derived from sisal corresponds to fragile, smooth fibres shown in figure 2c. The ACF fractured in a brittle failure and molten cross-link mode. The surface of the ACF was smooth on account of strong melting power of $ZnCl_2$ to cellulose.

Treated with DAP, ACF had much fewer melted fibres and rougher surface. But we also observed lots of melting dots existed in ACFs before the concentration of DAP was less than 12% from a series of preliminary studies about thermal stabilization. When the concentration of DAP reached 20%, as shown in figure 2d, most ACFs separated from other fibres. The samples were easily dispersed in normal water. The rough surface was formed through the escape of oxygen and hydrogen containing compounds and followed by enrichment of carbon particles. With the increasing concentration of DAP, melted fibres were barely found in the samples (figure 2e), but more small-scale fragments attached to the surface of the fibres.

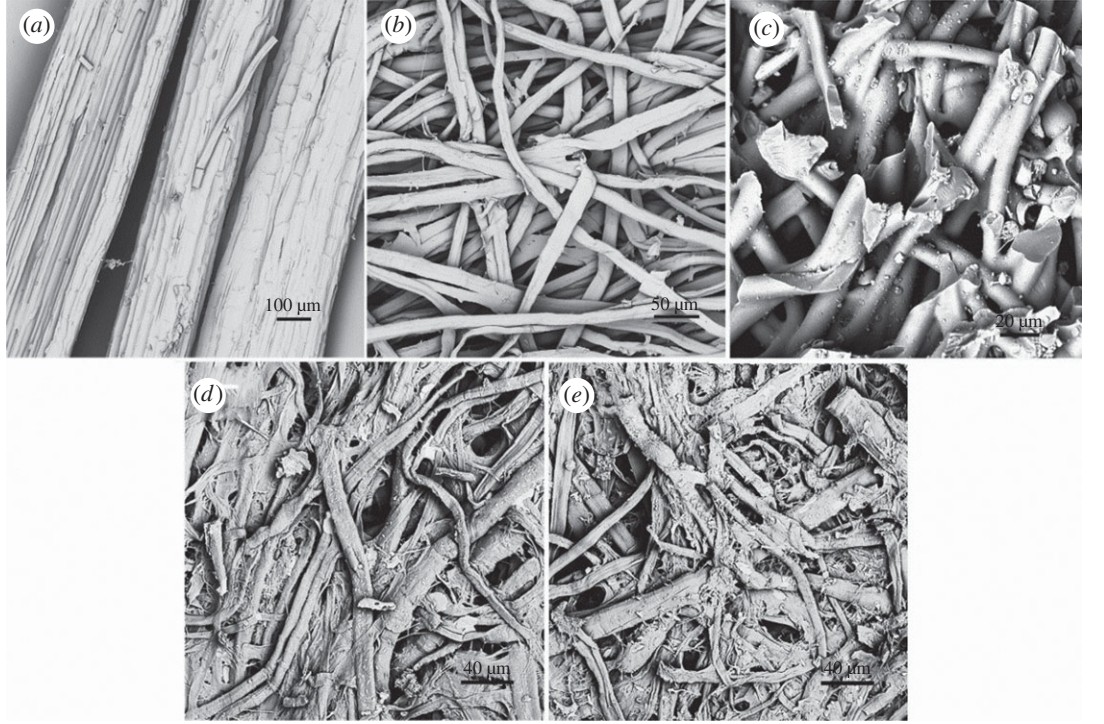

**Figure 2.** SEM images of sisal samples: (*a*) sisal xylem fibres; (*b*) separated sisal fibres; (*c*) SACF-DAP0; (*d*) SACF-DAP20; (*e*) SACF-DAP40.

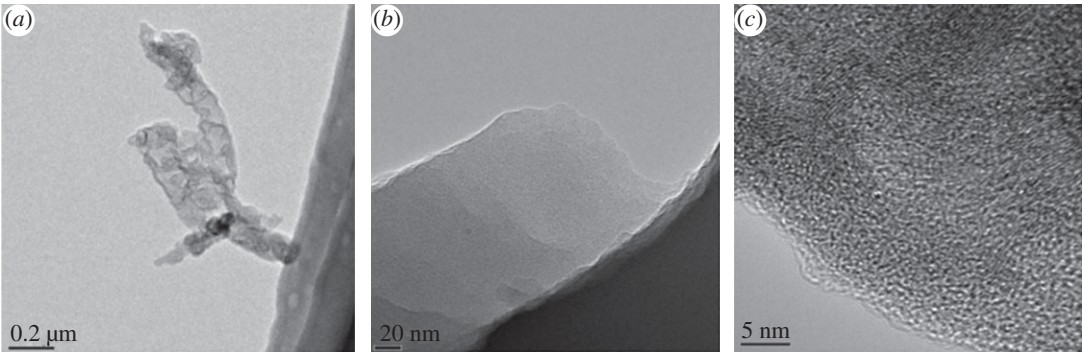

**Figure 3.** TEM images of SACF under different magnification: (*a*) 35 000×; (*b*) 800 000×; (*c*) 2 000 000×.

The TEM analyses of the SACF–DAP were conducted and the results are displayed in figure 3. Figure 3*a* represents one single ACF fibre with a flexible structure. The width of the fibre is approximately 180 nm. Structural details are shown in figure 3*b*. The edge of the ACF fibre has a layer structure with a mass of pores evenly distributed in it. The presence of the porous structure and the formation of graphite-like micro-crystal (0.34 nm) can clearly be seen in figure 3*c*, which can be due to high-temperature treatment. Porous structures are useful for facilitating the adsorption process. And graphitized structures shows the potential of high electrical conductivity, which can be used as electrode of supercapacitors [22].

## 3.3. Pore characteristics

For the investigation, adsorption isotherms of $N_2$ measured the structural properties of ACF samples. For comparison, the data were included of a commercial coconut shell-based GAC with advanced fuel adsorption performance widely applied in charcoal canisters. Figure 4 shows $N_2$ adsorption isotherms at 77 K of all samples to determine the effect of the DAP as thermal stabilizer used on the pore characteristics of ACF samples. The isotherms of samples had a classical type I-isotherm as IUPAC classified, characteristic of microporous materials. Relatively low pressure ($P/P^0 < 0.1$), the adsorption isotherms showed a steep curve which indicated large adsorption potential in micropore (width,

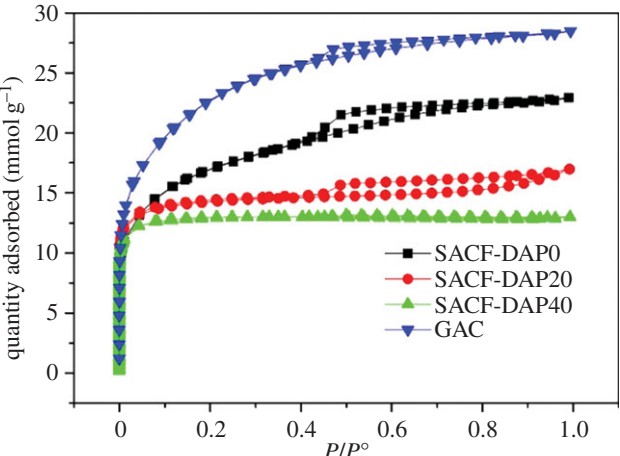

**Figure 4.** N$_2$ adsorption–desorption isotherms of SACF samples treated with different concentrations of DAP and commercial GAC.

**Table 2.** Pore Characteristics of SACF samples.

| sample | S$_{BET}$ (m$^2$g$^{-1}$) | S$_{MIC}$ (m$^2$g$^{-1}$) | S$_{MES}$ (m$^2$g$^{-1}$) | V$_{MIC}$ (cm$^3$g$^{-1}$) | W$_{MIC}$ (Å) |
|---|---|---|---|---|---|
| SACF-DAP0 | 1388.51 | 833.73 | 554.78 | 0.348 | 5.9 |
| SACF-DAP20 | 1228.96 | 1040.12 | 188.84 | 0.409 | 4.5 |
| SACF-DAP40 | 1137.68 | 1051.09 | 86.59 | 0.409 | 4.4 |
| GAC | 1841.29 | 1009.68 | 831.61 | 0.415 | 11.9 |

2 nm) adsorption area [23–25]. This form implies the existence of porous carbon contained substantial proportion micropores.

Pore characteristics of SACF samples, shown in table 2, were obtained by analysing the isotherms. Comparing with SACF-DAP0, SACF with thermal stabilization showed slightly lower surface area with a greater area of micropores, and the mean width of micropores also performed the same trend.

The amount of micropores in SACF-DAP40 was the relatively greater than other ACF samples. The result suggested that thermal stabilization treatment of DAP promotes forming of micropores and interferes with forming of mesopores (width between 2 and 50 nm) [26].

PSD of ACF samples is shown in figure 5. The results show that the size of multiple pores for the ACF sample is in the range of 0.5–2 nm, consistent with type I adsorption isotherm. Shown in magnification figure 5a, the volume of micropores increased; on the other hand, mesopores at range of 2–6 nm with DAP impregnation, more mesopores turned to micropores and the size of micropores became smaller.

## 3.4. Butane adsorption

The amount of micropores in SACF-DAP40 was greater than other ACF samples. The density strongly affects the calculated BWC, because the BWC is measured per 100 ml of carbon. In other words, comparing samples on a capacity basis, low-density materials have less carbon mass [19].

From a fundamental perspective, the BWC test method evaluated how much butane is adsorbed on AC and has characteristics similar to that of fuel. The BWC performance is based on the adsorption, desorption and repetition performance.

After feeding the sample with butane and nitrogen, desorption was repeated to determine the weight difference before and after the mixed gas was supplied. BWC outlined the porosity of the adsorbent. This is particularly valuable because these adsorbents usually have classic type I adsorption isotherm, which makes it difficult to analyse the PSD. Both micropores and mesopores are in samples that are suitable for butane adsorption. However, the ease of desorption of butane indicates mesoporosity because the butane adsorbed in the micropores is not easily desorbed.

The butane adsorption capacities of three ACF samples derived from sisal fibres and commercial GAC are shown in table 3. BWC of SACF-DAP20, which has much less BET specific surface area,

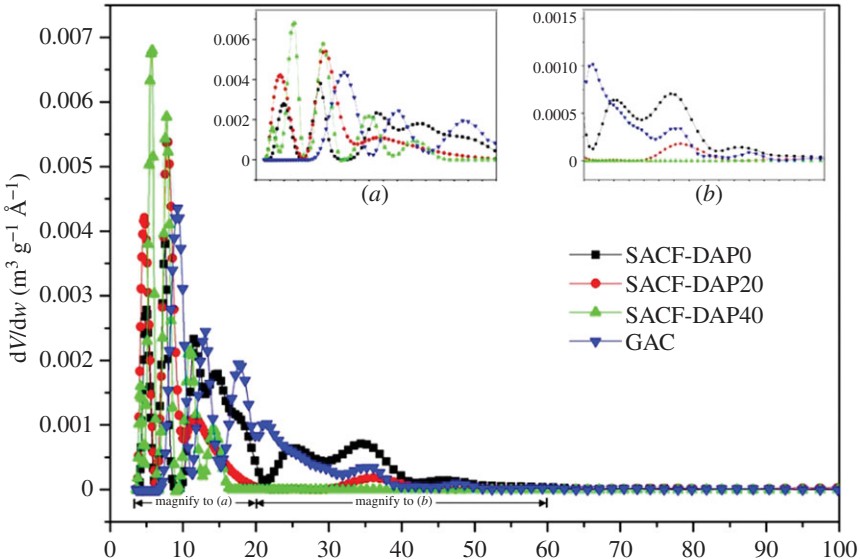

**Figure 5.** PSD of SACF samples: (*a*) magnification of 3–20 Å; (*b*) magnification of 20–60 Å.

**Table 3.** Butane working capacities of SACF samples and commercial GAC.

| sample | packing density (g l$^{-1}$) | BWC (WV, g 100 ml$^{-1}$) | BWC (WW, g 100 g$^{-1}$) |
|---|---|---|---|
| SACF-DAP0 | 496.3 | 12.8 | 25.8 |
| SACF-DAP20 | 428.6 | 13.7 | 32.0 |
| SACF-DAP40 | 446.9 | 13.2 | 29.5 |
| GAC | 498.2 | 13.3 | 26.7 |

shows the largest amount, which indicates being moderately treated with DAP could increase the BWC. This variation inferred thermal stabilization with DAP can effectively improve the porosity utilization of butane adsorption. But with the concentration of DAP increasing to 40%, the BWC declined, which indicates great reduction of mesopore could also affect BWC of ACFs. Comparison with commercial GAC samples (1841.29 m$^2$ g$^{-1}$), SACF-DAP20 (1228.96 m$^2$ g$^{-1}$) had much greater utilization of BET specific surface area. Considering the pore model of GAC reported in several articles, ACFs have more micropores in the surface of the fibres, which makes adsorbate more easily reach adsorption site.

Using photoionization detector, the two groups, SACF-DAP20 and GAC, recorded butane leaking concentration through the outlet end. The breakthrough mass of butane was recorded and plotted in the following figures. The absolute value of the slope reflects the velocity of butane adsorption/desorption. And the breakthrough point reflects the adsorption capacity of flowing adsorbate [27,28].

Figure 6 shows a comparison of two samples' adsorption breakthrough curves. In the first two cycles, SACF-DAP20 and commercial GAC were reaching 50% breakthrough point at the same time, but on the other hand, SACF-DAP20 was reaching saturation of adsorption capacity nearly 2 min faster than GAC. In the last three cycles, the adsorption rate, the slope, of GAC decreased heavily.

As the adsorption process was running, more pores, especially mesopores, were stopped by adsorbate. When some certain mesopores stopped, the micropores linked to these mesopores may have been unable to adsorb butane easily. SACF shows similar adsorption rate at five cycles with different breakthrough points, which means some pores were stopped in each cycle and pores stopping did not affect the adsorption of other pores. Adsorption breakthrough curves of two samples showed most adsorption sites in micropores, in SACF sample, were closer to the surface, which supported a more effective adsorption process.

Desorption process is shown in figure 7. Each cycle of both samples lasted less than 5 min. But SACF showed a more efficient desorption process (approximately 3 min) without obvious fluctuation, which also indicated the butane molecular desorbed more easily from SACF during desorption process. Desorption process reveals the service life of adsorbent. Higher speed of desorption process means more effective adsorption site existed in mesopores are stopped.

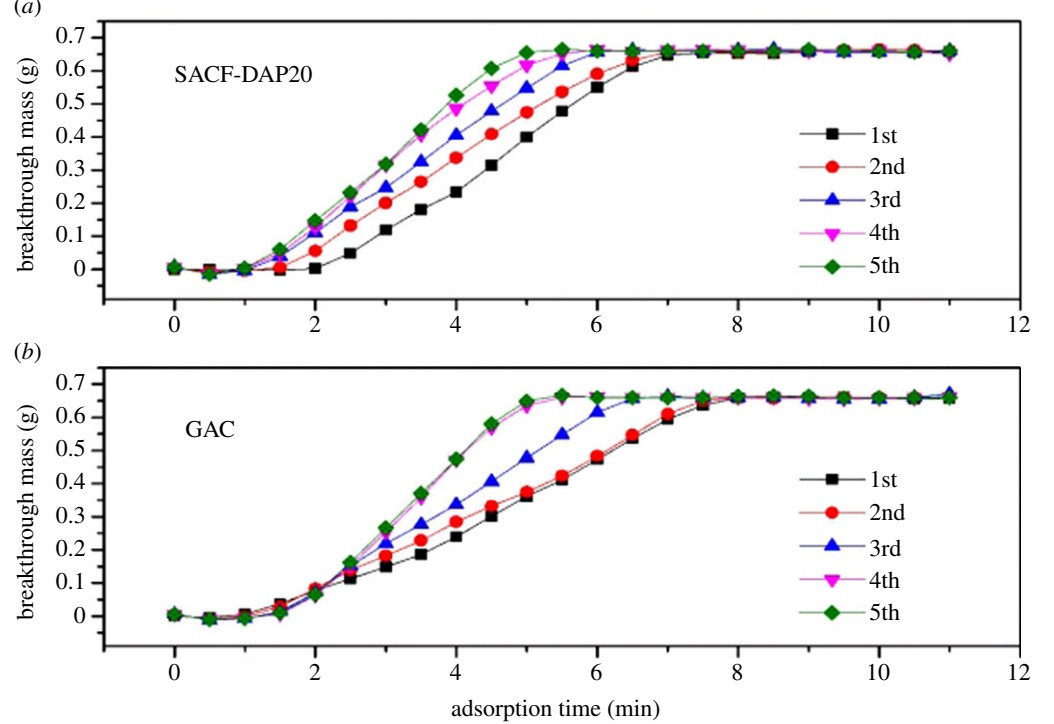

**Figure 6.** Five cycles of adsorption process of (*a*) SACF-DAP20 and (*b*) GAC.

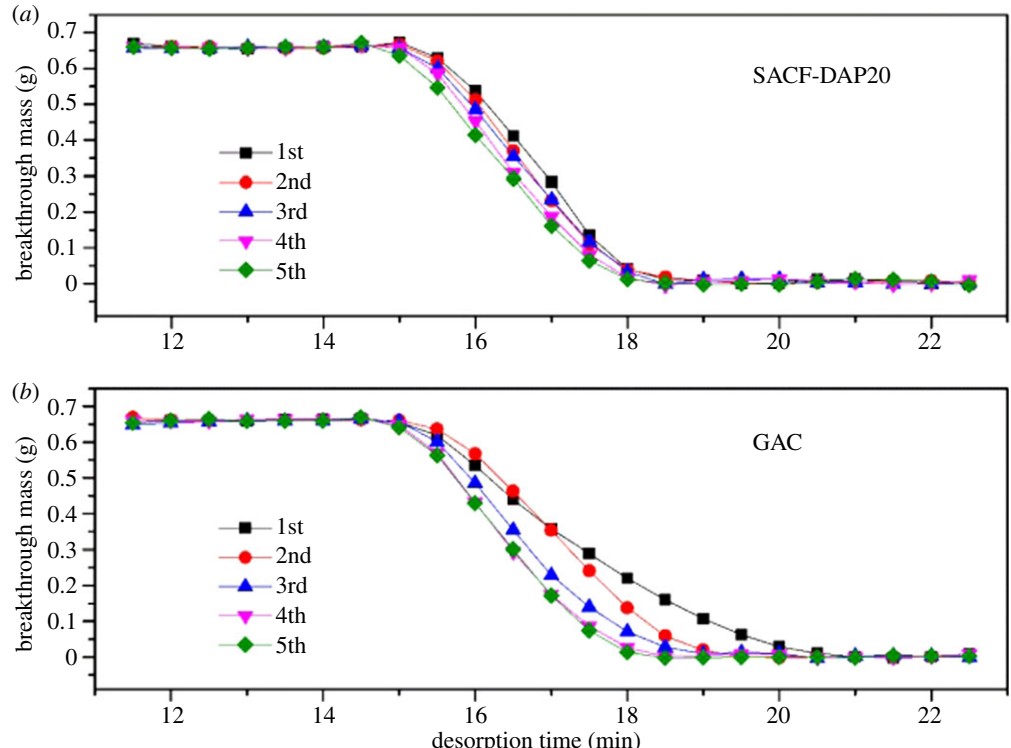

**Figure 7.** Five cycles of desorption process of (*a*) SACF-DAP20 and (*b*) GAC.

## 3.5. Kinetic study

To further study the kinetics of butane adsorption on the samples, two representative dynamic models, Thomas and Clark models, were chosen to analyse the breakthrough curves of butane adsorption.

**Table 4.** Parameters of Thomas model for butane adsorption on SACF-DAP20 and GAC.

| sample | cycle | $k_{th}$ (ml min$^{-1}$ mg$^{-1}$) | s.e. (ml min$^{-1}$ mg$^{-1}$) | $q_{th}$ (mg g$^{-1}$) | s.e. (mg g$^{-1}$) | $R^2$ |
|---|---|---|---|---|---|---|
| SACF-DAP20 | 1st | 0.375 | 0.014 | 291.705 | 2.355 | 0.997 |
| | 2nd | 0.349 | 0.015 | 255.034 | 2.825 | 0.996 |
| | 3rd | 0.382 | 0.018 | 225.504 | 2.893 | 0.995 |
| | 4th | 0.443 | 0.018 | 200.545 | 2.146 | 0.997 |
| | 5th | 0.484 | 0.021 | 193.102 | 2.119 | 0.997 |
| GAC | 1st | 0.276 | 0.011 | 258.727 | 3.010 | 0.997 |
| | 2nd | 0.266 | 0.014 | 246.607 | 4.004 | 0.996 |
| | 3rd | 0.348 | 0.018 | 215.164 | 3.045 | 0.995 |
| | 4th | 0.530 | 0.019 | 184.892 | 1.407 | 0.997 |
| | 5th | 0.535 | 0.023 | 182.288 | 1.612 | 0.997 |

Thomas adsorption model is a classical model for studying the dynamic adsorption breakthrough curve, which is based on the Langmuir adsorption theory and the second-order reaction kinetics model [27,28]. The equation of Thomas model is as follows:

$$\frac{C}{C_0} = \frac{1}{1 + \exp{[k_{th} C_0 ((q_{th} m / Q C_0) - t)]}},$$

(3.2)

where $C_0$ is initial adsorbate concentration (mg l$^{-1}$), $C$ is effluent adsorbate concentration (mg l$^{-1}$), $m$ is total dry weight of adsorbent (g), $Q$ is volumetric flow rate (ml min$^{-1}$) and $t$ is adsorption time (min). $k_{th}$ is rate constant of Thomas model (ml min$^{-1}$ mg$^{-1}$). The variations of this constant reflect the variation trend of the adsorption penetration rate of the sample. $q_{th}$ is saturated adsorption capacity of Thomas model (mg g$^{-1}$), and its variations reflects the trend of adsorption capacity of adsorbent.

Clark model combines mass transfer equation and Freundlich equation, which can be used to explain and predict the adsorption behaviour of adsorbate on adsorbent by analysing the adsorption breakthrough curve with nonlinear regression analysis method [29,30]. The equation of Clark model is as follows:

$$\frac{C}{C_0} = \left(\frac{1}{1 + A e^{-rt}}\right)^{1/(n-1)}.$$

(3.3)

$A$ is dimensionless Clark constant, which is related to the adsorption efficiency and intensity. $r$ is Clark constant (min$^{-1}$), which is inversely related to the breakthrough time and mass transmit rate. $n$ is the Freundlich constant, which is obtained ($n = 1.95$) from the previous static adsorption experiments.

We used the Thomas adsorption model to fit the adsorption breakthrough curve in figure 6, and the data obtained are shown in table 4. The correlation coefficients $R^2$ are all greater than 0.995, and the standard deviation is much smaller than the calculated value, which indicates that the model has a good fit to the data and has high reliability.

After five cycles, the $k_{th}$ of SACF-DAP20 increased by 30.5%, the $k_{th}$ of GAC increased by 93.8%, and the overall rate of SACF was slightly faster than GAC. The $q_{th}$ of SACF-DAP20 decreased by 33.8%, and the $q_{th}$ of GAC decreased by 29.5%. The adsorption capacity of each cycle of SACF-DAP20 was greater than that of GAC. It can be seen that although the specific surface area of GAC is relatively large, the overall butane adsorption capacity of SACF is stronger than GAC.

However, a careful comparison of the numerical changes can still find the deficiencies of SACF-DAP20 compared with GAC. At the 4th cycle, the $k_{th}$ of GAC surpassed SACF-DAP20. At this time, the proportion of pores that can be regenerated increased, so it can be inferred that this part of GAC has a higher adsorption rate on butane. With each cycle, the decrease in the adsorption capacity of SACF is greater, which indicates that the adsorption potential for butane of SACF-DAP20 is greater, and ordinary gas purging cannot make butane desorb well from SACF-DAP20.

The Clark model was used to perform nonlinear regression analysis on the adsorption breakthrough curve in figure 6, and the results obtained are shown in table 5. Overall, Clark model fits well and is suitable for analysing the adsorption breakthrough behaviour of butane on the samples. The

**Table 5.** Parameters of Clark model for butane adsorption on SACF-DAP20 and GAC.

| sample | cycle | $A$ | s.e. | $r$ (min$^{-1}$) | s.e. (mg g$^{-1}$) | $R^2$ |
|---|---|---|---|---|---|---|
| SACF-DAP20 | 1st | 155.970 | 31.282 | 1.128 | 0.043 | 0.997 |
| | 2nd | 60.610 | 10.979 | 1.051 | 0.044 | 0.995 |
| | 3rd | 52.695 | 10.449 | 1.149 | 0.055 | 0.995 |
| | 4th | 60.383 | 10.497 | 1.336 | 0.054 | 0.997 |
| | 5th | 74.614 | 14.942 | 1.457 | 0.065 | 0.996 |
| GAC | 1st | 45.424 | 7.749 | 0.830 | 0.035 | 0.994 |
| | 2nd | 33.347 | 6.692 | 0.802 | 0.043 | 0.991 |
| | 3rd | 54.890 | 12.261 | 1.047 | 0.055 | 0.994 |
| | 4th | 194.460 | 39.835 | 1.596 | 0.060 | 0.998 |
| | 5th | 188.612 | 44.495 | 1.609 | 0.070 | 0.997 |

dimensionless Clark constant $A$ of SACF decreased with the increase of cycle times, which may be due to the incomplete desorption of some pores with high adsorption potential energy. The $A$ of GAC showed an upward trend, which may be due to the failure of some mesopores and the remaining pores have strong adsorption capacity. But the standard error of $A$ is relatively large, which may cause errors in the analysis. The Clark constant $r$ increased in the breakthrough curves of the two adsorbents, which indicated that the adsorption breakthrough time was shorter and the mass transfer rate was faster when more pores failed. However, the $r$ of GAC was nearly doubled after four circuits, which indicated more pores failed in GAC sample. And the adsorption performance of SACF in circuits was more stable.

## 4. Conclusion

An efficient way was shown to turn agricultural waste to high value-added products. With thermal stabilization of DAP impregnation, we produced a sisal-based ACF with fibres separated. Based on former results and discussions, the conclusions are drawn:

— During the period of SACF's manufacturing with DAP impregnation, the yield of SACF increased two to three times and the fibre shape still remained during high-temperature (750°C) activation. The type of SACF has steady fibre shape even after defibreing.
— SACF-DAP0 had a little higher specific surface area than the SACF with DAP impregnation. It indicated that DAP had a slight protective effect against pore forming. But DAP also protected micropores from enlarging to mesopores or macropores and increased the proportion of micropore specific surface area which would be more important for fuel adsorption.
— BWC of the samples, which indicates the ability of fuel adsorption, also showed that proper thermal stabilization with DAP can improve fuel adsorption of SACF. Breakthrough curves of butane adsorption showed excellent fit to Thomas model and Clark model, which indicated both dynamic models were suitable for the description of butane adsorption. The breakthrough curves and parameters of Thomas model showed the SACF made with special thermal stabilization had better butane adsorption/desorption performance comparing with the commercial GAC. And parameters of Clark model indicated failure process of pores in circuit and SACF had a more stable adsorption performance to butane.

The SACF was prepared for volatile fuel adsorption, which was considered as raw material of functional paper or felt. The SACF productions would applied in intake tube, canister or air-filter assembly, which required certain strength to support customized cut and bending. This SACF can also be modified with special properties like anti-bacteria, precious metal reduction, hydrogen storage, etc.

Data accessibility. The datasets supporting this article have been uploaded as part of the electronic supplementary material.
Authors' contributions. Z.D. and J.H. designed the study. Z.D. carried out the laboratory work, data analysis and manuscript draft. H.L. coordinated the study and helped draft the manuscript. All the authors gave their final approval for publication.

Competing interests. We have no competing interests.

Funding. This research was funded by National Key R&D Program of China (grant no. 2017YFB0308000) and Guangdong Natural Science Funds for Distinguished Young Scholar (grant no. 2019B151502043).

Acknowledgements. The authors acknowledge Zhuzhou Times New Material Technology Co., Ltd for providing the sisal tissue for this research.

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
