## [Reviewer comments · Royal Society Open Science]

Review History

RSOS-200248.R0 (Original submission)

Review form: Reviewer 1

Is the manuscript scientifically sound in its present form?

No

Are the interpretations and conclusions justified by the results?

No

Is the language acceptable?

Yes

Do you have any ethical concerns with this paper?

Yes

Have you any concerns about statistical analyses in this paper?

No

Recommendation?

Reject

Comments to the Author(s)

The authors shows, activated carbon fibers (ACFs) are considered as the next generation of activated carbon products.

They say that in this paper, a sisal-based activated carbon fiber (SACF) material was prepared from sisal wastes with a unique thermal stabilization treatment to maintain fibrous shapes of SACFs while dispersing in solutions, and the SACFs were prepared as raw fiber materials for fuel evaporation emissions controlling products. The authors shows that with the experimental results of N₂ adsorption showed that SACF has a typical I-type adsorption isotherm, with specific surface area of SACF samples of around 1200 m²/g, and mainly microporous pore structure. Compared with commercial samples (specific surface area, 1841.3 m²/g), the butane working capacities (BWC) of SACF for advanced fuel evaporation emissions controlling product was 0.4g/100mL higher. Furthermore, Thomas dynamic model was applied to adsorption breakthrough data, which showed excellent fit. And it indicated from the adsorption breakthrough curves and parameters of Thomas model that the SACF has better performance in fuel adsorption and desorption process than the commercial samples.

The work, despite being registered, does not contribute anything original to the state of the art. Taking fibers from the agave (sisal) to obtain a new material is not sufficient, especially when the authors do not clearly demonstrate what is the progress and the possible production costs if they can be used massively.

I must highlight several aspects that the authors do not clearly present, so I must be against their publication.

1. In the introduction to work, the central theme is not clear and is not presented because sisal is important; for example how many tons of sisal does China produce? This is a Mexican waste product; the same question, how much is discarded?
2. What are the mechanical properties before and after this material? The authors do not discuss this aspect.
3. The methodology is not clear. The authors should be clearer about it.
4. In the analysis of results, the authors show the little knowledge they have on the subject. Just look at the reports of the specific area BET: how they report the area with undecimal! This is an error that is commonly passed on by al to the specialized literature, unfortunately. What kind of equipment can deliver this result like this? None.
4. The results of the textural part is very regular. The authors do not make a careful analysis of porosity: why do they not use models like NLDFT and QSDFT?
5. In this type of materials it is necessary to simulate the adsorption isotherms using computational models such as molecular dynamics, Monte Carlo, etc.
6. Model of the adsorption of butane and its interpretation is really regular.
7. This study lacked a kinetic and thermodynamic study.
8. A study using adsorption calorimetry is necessary.

Review form: Reviewer 2

Is the manuscript scientifically sound in its present form?

Yes

Are the interpretations and conclusions justified by the results?

Yes

Is the language acceptable?

Yes

Do you have any ethical concerns with this paper?

No

Have you any concerns about statistical analyses in this paper?

No

Recommendation?

Accept with minor revision (please list in comments)

Comments to the Author(s)

The manuscript from the point of view of basic research is interesting, but I'm not sure about the practical application. What is the amount of these kind waste available in this area? Are there potential users and where?

Decision letter (RSOS-200248.R0)

15-Apr-2020

Dear Mr Deng:

Manuscript ID: RSOS-200248

Title: "Sisal xylem fiber based activated carbon fibers for fuel adsorption: Effect of thermal stabilization of DAP"

Thank you for submitting the above manuscript to Royal Society Open Science. Your paper was sent to reviewers and their comments are included at the bottom of this letter.

In view of the concerns raised by the reviewers, the manuscript has been rejected in its current form. However, a new manuscript may be submitted which takes into consideration these comments.

Please note that resubmitting your manuscript does not guarantee eventual acceptance, and that your resubmission will be subject to peer review before a decision is made.

Your resubmitted manuscript should be submitted by 13-Oct-2020. If you are unable to submit by this date please contact the Editorial Office.

On behalf of the Subject Editor Professor Anthony Stace and the Associate Editor Dr Ya-Wen Wang

REVIEWER(S) REPORTS:

Associate Editor Comments to Author ():

RSC Associate Editor:

Comments to the Author:

(There are no comments.)

RSC Subject Editor:

Comments to the Author:

(There are no comments.)

Reviewers' Comments to Author:

Reviewer: 1

Comments to the Author(s)

The authors shows, activated carbon fibers (ACFs) are considered as the next generation of activated carbon products.

They say that in this paper, a sisal-based activated carbon fiber (SACF) material was prepared from sisal wastes with a unique thermal stabilization treatment to maintain fibrous shapes of SACFs while dispersing in solutions, and the SACFs were prepared as raw fiber materials for fuel evaporation emissions controlling products. The authors shows that with the experimental results of N₂ adsorption showed that SACF has a typical I-type adsorption isotherm, with specific surface area of SACF samples of around 1200 m²/g, and mainly microporous pore structure. Compared with commercial samples (specific surface area, 1841.3 m²/g), the butane working capacities (BWC) of SACF for advanced fuel evaporation emissions controlling product was 0.4g/100mL higher. Furthermore, Thomas dynamic model was applied to adsorption breakthrough data, which showed excellent fit. And it indicated from the adsorption breakthrough curves and parameters of Thomas model that the SACF has better performance in fuel adsorption and desorption process than the commercial samples.

The work, despite being registered, does not contribute anything original to the state of the art. Taking fibers from the agave (sisal) to obtain a new material is not sufficient, especially when the authors do not clearly demonstrate what is the progress and the possible production costs if they can be used massively.

I must highlight several aspects that the authors do not clearly present, so I must be against their publication.

1. In the introduction to work, the central theme is not clear and is not presented because sisal is important; for example how many tons of sisal does China produce? This is a Mexican waste product; the same question, how much is discarded?
2. What are the mechanical properties before and after this material? The authors do not discuss this aspect.
3. The methodology is not clear. The authors should be clearer about it.
4. In the analysis of results, the authors show the little knowledge they have on the subject. Just look at the reports of the specific area BET: how they report the area with undecimal! This is an error that is commonly passed on by al to the specialized literature, unfortunately. What kind of equipment can deliver this result like this? None.
4. The results of the textural part is very regular. The authors do not make a careful analysis of porosity: why do they not use models like NLDFT and QSDFT?
5. In this type of materials it is necessary to simulate the adsorption isotherms using computational models such as molecular dynamics, Monte Carlo, etc.
6. Model of the adsorption of butane and its interpretation is really regular.
7. This study lacked a kinetic and thermodynamic study.
8. A study using adsorption calorimetry is necessary.

Reviewer: 2

Comments to the Author(s)

The manuscript from the point of view of basic research is interesting, but I'm not sure about the practical application. What is the amount of these kind waste available in this area? Are there potential users and where?

Author's Response to Decision Letter for (RSOS-200248.R0)

See Appendix A.

RSOS-200966.R0

Review form: Reviewer 2

Is the manuscript scientifically sound in its present form?

Yes

Are the interpretations and conclusions justified by the results?

Yes

Is the language acceptable?

Yes

Do you have any ethical concerns with this paper?

No

Have you any concerns about statistical analyses in this paper?

No

Recommendation?

Accept as is

Comments to the Author(s)

You have to emphasize the possible practical application of your research.

Review form: Reviewer 3

Is the manuscript scientifically sound in its present form?

No

Are the interpretations and conclusions justified by the results?

Yes

Is the language acceptable?

No

Do you have any ethical concerns with this paper?

No

Have you any concerns about statistical analyses in this paper?

No

Recommendation?

Accept with minor revision (please list in comments)

Comments to the Author(s)

Manuscript Number: RSOS-200966

Title: Sisal xylem fiber based activated carbon fibers for fuel adsorption: Effect of thermal stabilization of DAP.

The revised manuscript is improved a lot. The authors are responded for the queries raised by the reviewers, but still some corrections are to be needed before the acceptance.

1. The introduction part is still weak. There is no strong literature about plant and sisal fibers.

First give the brief about the plant fibers and give the short note about the sisal fibers. For information the authors may kindly refer the following articles:

i) Plant fibre based bio-composites: Sustainable and renewable green materials

Renewable and Sustainable Energy Reviews 2017, 79, 558-584

Mechanical property evaluation of sisal-jute-glass fiber reinforced polyester composites, Composites Part B: Engineering 2013, 48, 1-9.

2. Language of the manuscript need improvement. Do the language editing of the manuscript with the help of the native English speaker.

3. In my opinion, Fig. 2 is not necessary. This may be removed.
4. The explanation for the SEM images presented in Fig. 3 is not sufficient. Need more explanation.
5. Rewrite the conclusion part in bullet points.

Decision letter (RSOS-200966.R0)

Dear Mr Deng:

Title: Sisal xylem fiber based activated carbon fibers for fuel adsorption: Effect of thermal stabilization of DAP

Manuscript ID: RSOS-200966

Thank you for submitting the above manuscript to Royal Society Open Science. On behalf of the Editors and the Royal Society of Chemistry, I am pleased to inform you that your manuscript will be accepted for publication in Royal Society Open Science subject to minor revision in accordance with the referee suggestions. Please find the reviewers' comments at the end of this email.

The reviewers and handling editors have recommended publication, but also suggest some minor revisions to your manuscript. Therefore, I invite you to respond to the comments and revise your manuscript.

Because the schedule for publication is very tight, it is a condition of publication that you submit the revised version of your manuscript before 25-Jul-2020. Please note that the revision deadline will expire at 00.00am on this date. If you do not think you will be able to meet this date please let me know immediately.

- 1) A text file of the manuscript (tex, txt, rtf, docx or doc), references, tables (including captions) and figure captions. Do not upload a PDF as your "Main Document".
- 2) A separate electronic file of each figure (EPS or print-quality PDF preferred (either format should be produced directly from original creation package), or original software format)
- 3) Included a 100 word media summary of your paper when requested at submission. Please ensure you have entered correct contact details (email, institution and telephone) in your user account

4) Included the raw data to support the claims made in your paper. You can either include your data as electronic supplementary material or upload to a repository and include the relevant doi within your manuscript

5) All supplementary materials accompanying an accepted article will be treated as in their final form. Note that the Royal Society will neither edit nor typeset supplementary material and it will be hosted as provided. Please ensure that the supplementary material includes the paper details where possible (authors, article title, journal name).

Kind regards,

Dr Laura Smith
Publishing Editor, Journals

RSC Associate Editor
Comments to the Author:
(There are no comments.)

Reviewer comments to Author:
Reviewer: 3

Comments to the Author(s)
Manuscript Number: RSOS-200966

Title: Sisal xylem fiber based activated carbon fibers for fuel adsorption: Effect of thermal stabilization of DAP.

The revised manuscript is improved a lot. The authors are responded for the queries raised by the reviewers, but still some corrections are to be needed before the acceptance.

1. The introduction part is still weak. There is no strong literature about plant and sisal fibers. First give the brief about the plant fibers and give the short note about the sisal fibers. For information the authors may kindly refer the following articles:

i) Plant fibre based bio-composites: Sustainable and renewable green materials
Renewable and Sustainable Energy Reviews 2017, 79, 558-584

Mechanical property evaluation of sisal-jute-glass fiber reinforced polyester composites, Composites Part B: Engineering 2013, 48, 1-9.

2. Language of the manuscript need improvement. Do the language editing of the manuscript with the help of the native English speaker.
3. In my opinion, Fig. 2 is not necessary. This may be removed.
4. The explanation for the SEM images presented in Fig. 3 is not sufficient. Need more explanation.
5. Rewrite the conclusion part in bullet points.

Reviewer: 2

Comments to the Author(s)

You have to emphasize the possible practical application of your research.

Author's Response to Decision Letter for (RSOS-200966.R0)

See Appendix B.

Decision letter (RSOS-200966.R1)

Dear Mr Deng:

Title: Sisal xylem fiber based activated carbon fibers for fuel adsorption: Effect of thermal stabilization of DAP

Manuscript ID: RSOS-200966.R1

It is a pleasure to accept your manuscript in its current form for publication in Royal Society Open Science. The chemistry content of Royal Society Open Science is published in collaboration with the Royal Society of Chemistry.

RSC Associate Editor
Comments to the Author:
(There are no comments.)

Reviewer(s)' Comments to Author:

Appendix A

Dear Editors:

We would like to resubmit our manuscript (Manuscript ID RSOS-200248) entitled "Sisal xylem fiber based activated carbon fibers for fuel adsorption: Effect of thermal stabilization of DAP ", by Zhuo Deng, Jian Hu and Hailong Li, which we wish to be considered for publication in *Royal Society Open Science*.

Firstly, thank you for your kind letter and for reviewers' constructive comments concerning our article. I feel very sorry for quality of manuscript not meeting the requirement of the journal. But I'm very grateful that you were willing to give me the opportunity to resubmit the revised manuscript.

I've received the reviews from two experts from the decision letter. The comments are mind-blowing for me. The review reports detailed listed deficiencies accurately pointed the changes I have to make. I think these reviews really improved the quality of my manuscript and ongoing work. Now I revised my manuscript carefully refer to the reviews and some new thoughts. I also supplemented extra data to make our results convincing. In this revised version, all changes to the manuscript within the document were all highlighted.

I'll respond to reviews comments in detail as followed.

Reviewer: 1

The authors shows, activated carbon fibers (ACFs) are considered as the next generation of activated carbon products.

They say that in this paper, a sisal-based activated carbon fiber (SACF) material was prepared from sisal wastes with a unique thermal stabilization treatment to maintain fibrous shapes of SACFs while dispersing in solutions, and the SACFs were prepared as raw fiber materials for fuel evaporation emissions controlling products. The authors shows that with the experimental results of N₂ adsorption showed that SACF has a typical I-type adsorption isotherm, with specific surface area of SACF samples of around 1200 m²/g, and mainly microporous pore structure. Compared with commercial samples (specific surface area, 1841.3 m²/g), the butane working capacities (BWC) of SACF for advanced fuel evaporation emissions controlling product was

0.4g/100mL higher. Furthermore, Thomas dynamic model was applied to adsorption breakthrough data, which showed excellent fit. And it indicated from the adsorption breakthrough curves and parameters of Thomas model that the SACF has better performance in fuel adsorption and desorption process than the commercial samples.

The work, despite being registered, does not contribute anything original to the state of the art. Taking fibers from the agave (sisal) to obtain a new material is not sufficient, especially when the authors do not clearly demonstrate what is the progress and the possible production costs if they can be used massively.

I must highlight several aspects that the authors do not clearly present, so I must be against their publication.

Response: Thank you for your comments on our article. According to your suggestions, we have supplemented several data here and corrected several mistakes in our previous manuscript. Based on your comments, we have made extensive revisions to our previous manuscript. The detailed point-by-point responses are listed below.

1. In the introduction to work, the central theme is not clear and is not presented because sisal is important; for example how many tons of sisal does China produce? This is a Mexican waste product; the same question, how much is discarded?

Response: We collected latest data of sisal industry from industrial reports and WHO. Global sisal fiber production amounted to 280 kt/y in 2017. And China produced 38 kt in 2017. According to this paper (Broeren MLM, 2017 Life cycle assessment of sisal fiber – Exploring how local practices can influence environmental performance. J CLEAN PROD. 149, 818-827) which was already listed in the reference, the solid sisal fiber waste (SFW) fraction which will be discarded amounts to 14 t/t sisal fiber. Around 5% of SFW is the xylem fiber with production of 0.7t/t sisal fiber. I have added this part to the manuscript as suggested.

2. What are the mechanical properties before and after this material? The authors do not discuss this aspect.

Response: There are several papers studied the mechanical properties of natural fiber based ACF (Cho D, Kim JM, Song IS Hong I. 2011 Effect of alkali pre-treatment of jute on the formation of jute-based carbon fibers. MATER LETT. 65, 1492-1494.; .Ooi C, Cheah W, Sim Y, Pung S Yeoh F. 2017 Conversion and characterization of activated carbon fiber derived from palm empty fruit bunch waste and its kinetic study on urea adsorption. J ENVIRON MANAGE. 197, 199-205.), but I noticed the ACF they made is fiber bundle, not single fiber. In our study, I used alkaline pulping methods to separate the sisal fibers (7-14 μ m of diameter). The fibers we made have many kind with different mechanical properties. The mechanical properties of single fiber don't have representative. We usually tested the mechanical properties of paper made of these fibers. But in this study, our purpose is to discuss the adsorption performance of SACF as raw materials. Carbon paper made of SACF containing other research sections such as reinforcing fiber, adhesive, hot-pressing conditions will be our next stage of research.

3. The methodology is not clear. The authors should be clearer about it.

Response: I have revised preparation of raw materials with correct and detailed steps.

4. In the analysis of results, the authors show the little knowledge they have on the subject. Just look at the reports of the specific area BET: how they report the area with undecimal! This is an error that is commonly passed on by al to the specialized literature, unfortunately. What kind of equipment can deliver this result like this? None.

Response: I revised results with one more decimal place. In original reports provided by MicroActive software from Micromeritics, the result of BET specific area have four decimal places. I have consulted technical support of Micromeritics. The reports were set to have 3-4 decimal places in some results. They suggested that keeping two decimal places can eliminate the error of measurement. I also find many papers using BET specific area without decimal which were clearly not accurately.

4. The results of the textural part is very regular. The authors do not make a careful analysis of porosity: why do they not use models like NLDFT and QSDFT?

Response: We have used NLDFT to investigate the pore size distribution (PSD) and Fig.6 is the result of using NLDFT model. And in 4.3 Pore characteristics, we discussed the PSD analysed by NLDFT model.

5. In this type of materials it is necessary to simulate the adsorption isotherms using computational models such as molecular dynamics, Monte Carlo, etc.

Response: We totally agree the molecular dynamics is quite meaningful for investigating the adsorption isotherms. But in our study, we didn't use pressure swing adsorption to get the isotherm of butane adsorption on SACF. Instead, we used optimized BWC test system in standard (ASTM D5228-92) and got breakthrough curves to investigate the adsorption process. We found it commonly analysed breakthrough curves with dynamic models in these papers. (.Zhang X, Chen SBi HT. 2010 Application of wave propagation theory to adsorption breakthrough studies of toluene on activated carbon fiber beds. CARBON. 8 48, 2317-2326; .Ooi C, Cheah W, Sim Y, Pung S Yeoh F. 2017 Conversion and characterization of activated carbon fiber derived from palm empty fruit bunch waste and its kinetic study on urea adsorption. J ENVIRON MANAGE. 197, 199-205; .Han R, Wang Y, Zhao X, Wang YXie F. 2009 Adsorption of methylene blue by phoenix tree leaf powder in a fixed-bed column: experiments and prediction of breakthrough curves. DESALINATION. 245, 284-297)

6. Model of the adsorption of butane and its interpretation is really regular.

Response: We also agree using one kinetic model, Thomas model which is based on the Langmuir adsorption theory and the second-order reaction kinetics model, to analyse the breakthrough curve is regular. So we added another kinetic model, Clark model which is based on the use of a mass-transfer concept in combination with the Freundlich isotherm, for breakthrough curve analysis comparing with Thomas model results.

7. This study lacked a kinetic and thermodynamic study.

Response: We have added another kinetic model to enrich the kinetic study to the manuscript. We are also seeking the opportunity to do the adsorption microcalorimetry experiment in order to get more sophisticated data for the thermodynamic study. Investigating the thermodynamic study in detail is our next research plan.

8. A study using adsorption calorimetry is necessary.

Response: Indeed, the adsorption calorimetry is an advanced method to study adsorption behavior. We had planned to investigate the heat exchange in the adsorption of atomic or molecular species onto a surface and between the substrate and the surrounding environment. It's really a complicated research project which need detailed discussion. We are also seeking the opportunity to do the adsorption microcalorimetry experiment in order to get more sophisticated data for the thermodynamic study.

Reviewer: 2

Comments to the Author(s) The manuscript from the point of view of basic research is interesting, but I'm not sure about the practical application. What is the amount of these kind waste available in this area? Are there potential users and where?

Response: We feel great thanks for your professional review work on our manuscript. As you are concerned, there are several problems that need to be addressed. According to your nice suggestions, we have made extensive corrections to our previous manuscript, the detailed corrections are listed below.

Comments and Questions:

Line 20-23_ Please, clarify what is the unique thermal stabilization measure of this kind of fibers?

Response: $(\text{NH}_3)_2\text{HPO}_4$ (DAP) is usually used as a phosphate flame retardant or an activator of activated carbon preparation. In this study, DAP was turned to be efficient to keep the fiber shape during thermal process. And I described this part in first paragraph of 4.1 Burn-off.

Line 30_ Specify a place of local farm?

Response: I have added the detailed address of local farm as suggested.

Line 38-49_ In my opinion, these two sentences are unnecessary.

Response: I have deleted these two sentences as suggested. It's indeed no need to introduce the basic concept here.

In the Methods part of Manuscript you did not indicate that you analysed samples by TEM technique (type of device, recording conditions ...)

Response: I have written this part in first paragraph of 3.3 Activated carbon fibers Characterization. The type of device is JEM-2200FS, JEOL Japan. And I added the recording conditions.

Please, explain how you got separate fibers, which is shown in Figure 3b?

Response: I used alkaline pulping methods to separate the sisal fibers. It's classic pulping process. Heated in a digester, the pectin and lignin, binder of fibers, in fiber bundle will be massively dissolved in alkaline solution. And the fibers will be separated as shown in Figure 3b. And I added this description in the manuscript.

What is the amount of these kind waste available in the area where you used this type of fiber?

Response: We collected some data of sisal industry from industrial reports and WHO. Global sisal fiber production amounted to 280 kt/y in 2017. And China produced 38 kt in 2017. According to this paper (Broeren MLM, 2017 Life cycle assessment of sisal fiber – Exploring how local practices can influence environmental performance. J CLEAN PROD. 149, 818-827) which was already listed in the reference, the solid sisal

fiber waste (SFW) fraction which will be discarded amounts to 14 t/t sisal fiber. Around 5% of SFW is the xylem fiber with production of 0.7t/t sisal fiber. I have added this part to the manuscript.

Why did you decide to test adsorption of butane? It should be emphasized as one of the aims of your manuscript.

Response: The SACF is prepared as materials of hydrocarbon adsorber applied in cars. The butane working capacity (BWC) is an essential indicator of this type of product.

Which commercial products are you talking about at the end of the Conclusion?

Response: The SACF we prepared was raw material of carbon paper. This type of paper was made to compete with ACS series production from Ingevity Corporation. ACS series, which dominated the market for years, are a type of activated carbon paper applied on the inlet of cars to adsorb the hydrocarbon.

We appreciate for Editors and Reviewers' warm work earnestly, and hope that the correction will meet with approval. Once again, thank you very much for your comments and suggestions.

Sincerely,

Zhuo Deng

E-mail: jackdz123@sina.com

Corresponding author:

Name: Hailong Li

E-mail: felhl@scut.edu.cn

Appendix B

Dear Editors:

It's really glad to receive the decision letter this time. We are honored to get your positive decision. And we sincerely thank the editors and all of the reviewers for your constructive comments and immensely helpful suggestions. We have highlighted the changes made in the manuscript. We also provide point-to-point reply to each comment raised. We have tried our best to address all of the issues raised and hope that our revised manuscript can meet the expectations for publication.

I received comments from two reviewers in Email. I'll respond to the comments in detail as followed.

Reviewer: 1

Comments to the Author(s)

Manuscript Number: RSOS-200966

Title: Sisal xylem fiber based activated carbon fibers for fuel adsorption: Effect of thermal stabilization of DAP.

The revised manuscript is improved a lot. The authors are responded for the queries raised by the reviewers, but still some corrections are to be needed before the acceptance.

1. The introduction part is still weak. There is no strong literature about plant and sisal fibers. First give the brief about the plant fibers and give the short note about the sisal fibers. For information the authors may kindly refer the following articles:

i) Plant fibre based bio-composites: Sustainable and renewable green materials

Renewable and Sustainable Energy Reviews 2017, 79, 558-584

Mechanical property evaluation of sisal-jute-glass fiber reinforced polyester composites, Composites Part B: Engineering 2013, 48, 1-9.

Response 1: I noticed that I was using too many words on the necessity of using sisal xylem fibers as the raw materials. Lacking introduction of the properties of sisal fibers. Now I have revised this part to the manuscript as suggested and added these two articles in the References.

2. Language of the manuscript need improvement. Do the language editing of the manuscript with the help of the native English speaker.

Response 2: I've one of my American friends to help with the language of this paper. If there's any presentation may lead to misunderstanding, please don't hesitate to point out. I'll revise that as suggested.

3. In my opinion, Fig. 2 is not necessary. This may be removed.

Response 3: I have removed Fig.2 as suggested.

4. The explanation for the SEM images presented in Fig. 3 is not sufficient. Need more explanation.

Response 4: I have added more explanation of the SEM images.

5. Rewrite the conclusion part in bullet points.

Response 5: I have revised this part as suggested.

Reviewer: 2

Comments to the Author(s)

You have to emphasize the possible practical application of your research.

Response: I have added more explanation about the application in Conclusion.

We appreciate for Editors and Reviewers' warm work earnestly. Once again, thank you very much for your comments and suggestions.

Sincerely,

Zhuo Deng

E-mail: jackdz123@sina.com

Corresponding author:

Name: Hailong Li

E-mail: felhl@scut.edu.cn